# Extended spectrum β-lactamase (ESβL)-producing *E. coli* causing urinary tract infection among pregnant women and pediatric patients in public hospitals in northern Jordan

**Waleed Al Momani**[1], **Ayah Elayan**[2], **Rama Al Titi**[2], **Ismail Malkawi**[3], **Leen Al momani**[4], **Mohammad Al-Magableh**[2]*

1 Department of Basic Medical Sciences, Faculty of Medicine, Yarmouk University, P.O Box Irbid, Jordan, 2 Department of Pediatrics, Family Medicine and Obstetrics and Gynecology, Faculty of Medicine, Yarmouk University, Irbid, Jordan, 3 Department of Basic Veterinary Sciences, Faculty of Veterinary Medicine, Jordan University of Science and Technology, Ar-Ramtha, Jordan, 4 Faculty of Medicine, Jordan University of Science and Technology, Ar-Ramtha, Jordan

* mohammad.magableh@yu.edu.jo, md242003@yahoo.com

## Abstract

### Background

Urinary tract infections (UTI's) imply the invasion and multiplication of bacteria in the individual's urinary tract, the incidence of UTI's is mainly determined by sex and age being the highest in pregnant females. *E. coli* is the most frequently encountered causative agent of UTI's and recent data depict an increase in antimicrobial resistance.

### Objectives

This study aims to identify the most common pathogens associated with UTI's in pregnant women and children and to reveal the resistance patterns, and effective treatments and to detect resistance genes in *E. coli*.

### Methods

This study was conducted in public-teaching Hospitals in northern Jordan during the period from the 1st of March 2024 to the 5th of April 2024. Urine samples were collected in a sterile urine container or using a special tube for pediatric samples. Each sample was cultivated on MacConkey, blood and mannitol salt agar plates, all plates were incubated aerobically at 37°C for 24 hours and identified using standard microbiological protocols. All isolates were tested for antimicrobial susceptibility. Polymerase chain reaction was performed to confirm all *E. coli* isolates by the detection of universal stress protein (uspA) gene and to detect ESβL genes TEM, SHV CTX-M.

**Data availability statement:** All relevant data are within the paper and its Supporting Information.

**Funding:** The author(s) received no specific funding for this work.

**Competing interests:** The authors have declared that no competing interests exist.

## Results

Eight bacterial species were identified, 75% (6/8) Gram Negative, 25% (2/8) Gram Positive). *Escherichia coli* was the most frequently isolated species 71.4% (50/70) followed by *Klebsiella pneumoniae* 10% (7/70), *Enterobacter cloacae* 5.7% (4/70, *Proteus mirabilis* 4.3% (3/70), *Enterococcus faecalis* 2.9% (3/70), *Acinetobacter baumannii* 2.9% (3/70), *Lelliottia amnigena* 1.4% (1/70) and *Streptococcus agalactiae* 1.4% (1/70). All 50 biochemically confirmed *E. coli* isolates were also confirmed using uspA gene. Three resistance genes TEM, SHV and CTX-M were detected using polymerase chain reaction.

## Conclusion

*E. coli* was the most frequently isolated species followed by *K. pneumoniae*, this result may indicate that *K. pneumoniae* is becoming increasingly common as an etiological agent of UTI. A high resistance toward Amoxicillin- Clavulanic Acid was found among *E. coli* isolates, while multiple antibiotic/drug resistance was 62%. These results highlight the need to adhere to the therapeutic guidelines in treating UTI's and other infections and not overuse antibiotics to prevent the emergence of multidrug-resistant bacteria.

## Introduction

Urinary tract infections (UTI's) imply the invasion and multiplication of bacteria in the individual's urinary tract [1], it can be further classified based on the clinical case presentation and diagnosis into upper or lower UTI. Higher risk UTI is associated with high fever, sepsis, isolation of organisms other than non-extended spectrum-lactamase (ESβL)-producing *Escherichia coli (E. coli)*, acute kidney damage and/or detection of abdominal or vesical masses [2].

Incidence of UTI's is mainly determined by age and sex; a high incidence occurs in males up to 6 months of age and in females up to 1 year of age [2]. UTI occurs most frequently in 16–35 years old females, up to 10% get an annual infection and 40–60% got infected at some point in their life [3]. Furthermore, during pregnancy, the risk of developing pyelonephritis is increased by 20% [4].

Urine bacterial cultures revealed that *E. coli* is the most common causative agent of UTI's responsible for 60 to 80% of cases. Previous use of antimicrobials or the presence of genital tract abnormalities increases the occurrence of *Proteus mirabilis* (6 to 10%), *Klebsiella pneumoniae* (3 to 5%) and to a lesser extent *Enterobacter cloacae*, *Citrobacter Serratia* and *Morganella morganii* and *Enterococcus faecalis* [2]. To our best knowledge, this is the first study to investigate the common pathogens associated with UTI's in pregnant women and children in northern Jordan.

*Escherichia coli* is a Gram-negative, facultative anaerobic, rod-shaped, coliform of the genus *Escherichia* commonly found in warm-blooded organisms' lower intestine [5]. Most *E. coli* strains are non-pathogenic. However, serotypes such as EPEC, ETEC etc. can cause serious food poisoning in their hosts and are occasionally responsible for food contamination incidents that prompt product recalls [6].

Treatment options for UTI's include first-line antimicrobials, for example (Nitrofurantoin, Trimethoprim-Sulfamethoxazole (TMP-SMX), Pivmecillinam, and Fosfomycin Tromethamine) and two alternative agents (β-lactams and Fluoroquinolones) but due to the widespread of antibiotic-resistance mechanisms treatment becomes increasingly difficult [7].

ESβL *E. coli* degrades oxyimino-cephalosporin, which is used when the isolate resists penicillin, resistance to Cephalosporin is most serious since the group is used to treat highly

pathogenic human infections. ESβL is mainly represented by three genes, TEM, SHV and CTX-M, bacteria of CTX-M type have become a major cause of human infections [8,9,10].

This study aims to identify the most common pathogens associated with UTI infections in pregnant women and children in northern Jordan and to reveal the resistance patterns and the effective treatments and to detect resistance genes in *E. coli.*

## Methods

### Ethical Approval

The study protocol was approved by the director of Al-Bashir Hospitals Administration/ Chairman of the Scientific Research Ethics Committee No. M.B.A./ Ethics Committee/17513/ dated 29/10/2019 and approved by the Institutional Review Board (IRB) at Yarmouk University in Jordan, approval number (IRB/2024/118).

### Study design and period

A hospital-based prospective study was carried out in public-teaching Hospitals in northern Jordan with a total capacity of 400 beds (Badieah and Rahma Hospitals). 70 urine samples were collected from pediatrics and female outpatients. The study was carried out during the period from the 1st of March 2024 to the 5th of April 2024 and all analysis was done on Microsoft Excel.

### Study setting and population

Each participant gave written informed consent, and parental consent was obtained for children involved in this study. All patients (pregnant women and children) who met the criteria of getting UTI's (had symptoms and culture resulted in bacterial growth) were included in the study, seventy UTI cases were covered in this study.

### Samples collection and bacterial isolation

Samples were collected from each patient using the midstream clean-catch technique [11], each sample was contained in a sterile urine cup, placed in an ice box and transported to the lab within an hour. Upon arrival, each sample was streaked on MacConkey, mannitol salt agar and blood agar plates, all plates were incubated aerobically at 37°C for 24 hours. After incubation, colonies on both media were checked for morphology, sugar fermentation and hemolysis (lactose and mannitol), further cultivation was made to get pure cultures. Further biochemical confirmation of bacteria was done using the VITEK 2 system (Biomerieux, France) after identification, all bacterial isolates were preserved in glycerol and kept at −80c for further testing.

### Antimicrobial susceptibility testing

All isolates were tested for antimicrobial susceptibility using the disk diffusion method according to the Clinical and Laboratory Standards Institute (CLSI) guidelines document M100-Ed34 [12], the antimicrobials used were Ampicillin (AMP, 10μg), Norfloxacin (NOR, 10μg), Erythromycin (E, 15μg), Chloramphenicol (C, 30μg), Gentamicin (GEN, 10μg), Tetracycline (T, 30μg), Vancomycin (VA, 30μg), Amoxicillin/Clavulanic Acid (AMC, 20/10μg), Cefepime (FEP, 30μg), Penicillin (P, 10μg), Methicillin (ME, 2μg), Trimethoprim/Sulfamethoxazole (SXT, 1.25/23.75μg), Oxacillin (OX, 1μg), Ciprofloxacin (CIP, 5μg) e. Briefly, microorganisms were suspended in normal saline (0.9% NaCl) to the turbidity of 0.5 McFarland tube. A swab of the cell suspension was then spread on the surface of Mueller-Hinton Agar plate and left for air drying for maximum of 10 min at room temperature before antibiotic

disks were placed onto the agar. The agar plates were then incubated at 35°C for 18–24 hours. Susceptibility and resistance were determined after measuring the zone of inhibition.

## Molecular confirmation and detection of resistance genes

Polymerase chain reaction (PCR) was used to confirm all *E. coli* isolates by the detection of universal stress protein (*uspA*) gene and to detect ESβL genes TEM, SHV and CTX-M. All primers and reaction conditions are presented in Table 1.

Uniplex PCR reactions were prepared for each gene in a total volume of 20 µl which contained 4 µl template DNA, 4 µl Firepol 5x master mix (Solis Biodyne, Estonia), 0.6 µl of each primer and up to 20 µl nuclease free water.

## Results

### Bacterial isolates

Seventy bacterial isolates were identified in this study; the rate of bacterial isolation was 40% among adults and 60% among pediatrics. The most frequent bacterial species isolated from UTI patients was *E. coli* (50/70, 71.4%) Fig 1, followed by *Klebsiella pneumoniae* (7/70, 10%),

**Table 1. Primers used in this study along with reaction conditions and expected product size.**

| Amplified gene | Primer | Cycles, Product Size | Ref |
|---|---|---|---|
| *uspA* | F:CCGATACGCTGCCAATCAGT | 95°C for 5mins followed by 35 cycles of | [8] |
| | R:ACGCAGACCGTAGGCCAGAT | 95 °C for 30s, 56°C for 30s and 72°C for 45s and finally 72°C for 10mins, 884 bp | |
| *TEM* | F: ATAAAATTCTTGAAGACGAAA | 95°C for 5mins followed by 35 cycles of | [9] |
| | R:GACAGTTACCAATGCTTAATCA | 95 °C for 30s, 55°C for 30s and 72°C for 45s and finally 72°C for 10mins, 1080 bp | |
| *SHV* | F: GGGTTATTCTTATTTGTCGC | 95°C for 5mins followed by 35 cycles of | |
| | R: TTAGCGTTGCCAGTGCTC | 95 °C for 30s, 56°C for 30s and 72°C for 45s and finally 72°C for 10mins, 930 bp | |
| *CTX-M* | F: CGCTTTGCGATGTGCAG | 95°C for 5mins followed by 35 cycles of | |
| | R: ACCGCGATATCGTTGGT | 95 °C for 30s, 60°C for 30s and 72°C for 45s and finally 72°C for 10mins, 550 bp | |

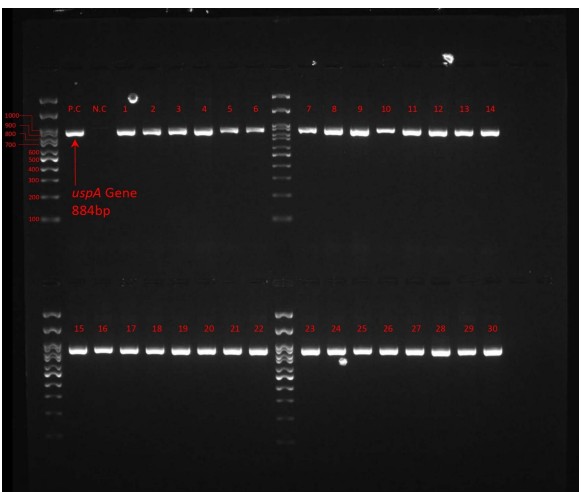

**Fig 1. Electrophoresis of *uspA* gene, 100 bp Ladder, Lane P.C: Positive Control, Lane N.C: Negative Control, Lane 1-30: Isolates.**

*Enterobacter cloacae* (4/70, 5.7%), *Proteus Mirabilis* (3/70, 4.3%), *Acinetobacter baumannii* (2/70, 2.9%) and *Enterococcus faecalis* (2/70, 2.9%) and *Streptococcus agalactiae* (1/70, 1.4%) and *Lelliottia amnigena* (1/70, 1.4%) as shown in Table 2.

## Antimicrobial susceptibility testing

Generally, resistance levels are high, they were the highest against Penicillin, Ampicillin, Erythromycin, Oxacillin, Methicillin and Vancomycin among all isolates. Susceptibility varied depending on the species, *E. coli* isolates were least resistant to Chloramphenicol, Cefepime, Norfloxacin and Gentamicin, while *K. pneumoniae* Isolates were least resistant to Chloramphenicol, Trimethoprim-Sulfa, Norfloxacin, Amoxicillin Clavulanic Acid and Gentamicin. Furthermore, *P. mirabilis* resistance was the least towards Chloramphenicol, Cefepime, Norfloxacin and Gentamicin. On the other hand, *E. cloacae* isolates were least resistant to Chloramphenicol, Tetracycline, Trimethoprim-Sulfa and Norfloxacin.

Two *E. faecalis* isolates resisted Cefepime, Vancomycin, Oxacillin, and Methicillin and one *S. agalactiae* isolate resisted Penicillin, Ampicillin, Chloramphenicol, Tetracycline, Cefepime, Vancomycin, Norfloxacin, Gentamicin, Erythromycin and Methicillin. Table 3–4.

Moreover, multidrug resistance (MDR) (defined as bacteria that are resistant to one or more classes of antibiotics) for each isolate and each group were above 0.2, the average MAR for all 70 isolates was 0.66 (66%), as shown in Table 5.

## Detection of resistance genes

Detection of ESβL gene revealed that among all *E. coli* isolates, $ESβL_{TEM}$ was the most frequent (52%, 26/50) followed by $ESβL_{CTX-M}$ (38%, 19/50), $ESβL_{SHV}$ was not detected in any of *E. coli* isolates. Fig 2. Figs 3–5 show the electrophoresis results for ESβL genes.

## Discussion

Urinary tract infections (UTI's) are encountered frequently in pregnant women [13], and it is one of the most common bacterial infections in children. In this study, the occurrence of UTI in children was higher than adults (60% and 40% respectively). Eight out of 100 children will experience at least one UTI between the ages of 1 month to 11 years [14]. Moreover, up to 30% of children and infants will go throw a recurrent infection 6 - 12 months after the initial UTI [15,16].

These studies' findings revealed the predominance of gram-negative bacteria as the causative agent of UTI, with *E. coli* being the most frequently isolated species (71.4%), similar

**Table 2. Isolation rates of each bacterial species among pregnant women and pediatric patients from this study.**

| Isolate | Adults (%) | Pediatrics (%) | Total (%) |
|---|---|---|---|
| *E. coli* | 18 (36%) | 32 (64%) | 50 (71.4%) |
| *K. pneumoniae* | 4 (57%) | 3 (43%) | 7 (10%) |
| *E. cloacae* | 2 (50%) | 2 (50%) | 4 (5.7%) |
| *P. Mirabilis* | 1 (33%) | 2 (67%) | 3 (4.3%) |
| *A. baumannii* | 0 (00%) | 2 (100%) | 2 (2.9%) |
| *E. faecalis* | 2 (100%) | 0 (00%) | 2 (2.9%) |
| *L. amnigena* | 0 (00%) | 1 (100%) | 1 (1.4%) |
| *S. agalactiae* | 1 (100%) | 0 (00%) | 1 (1.4%) |
| Total | 28 (40%) | 42 (60%) | 70 (100%) |

According to age, *E. coli* and *Proteus* were isolated from pediatrics more than adults, however, *Klebsiella pneumoniae* was isolated from adults more than pediatrics.

**Table 3. Prevalence and number of resistant isolates in this study.**

| Antimicrobial Agent | E. coli; 50 Isolates | | K. pneumoniae; 7 Isolates | | P. Mirabilis; 3 Isolates | | E. cloacae; 4 Isolates | |
|---|---|---|---|---|---|---|---|---|
| | Pregnant Women; 18 | Pediatrics; 32 | Pregnant Women; 4 | Pediatrics; 3 | Pregnant Women; 1 | Pediatrics; 2 | Pregnant Women; 2 | Pediatrics; 2 |
| Penicillin (P) | 18 (100%) | 32 (100%) | 4 (100%) | 3 (100%) | 1 (100%) | 2 (100%) | 2 (100%) | 2 (100%) |
| Ampicillin (AMP) | 16 (88.9%) | 30 (93.8%) | 4 (100%) | 3 (100%) | 1 (100%) | 2 (100%) | 2 (100%) | 2 (100%) |
| Chloramphenicol (C) | 2 (11.1%) | 5 (15.6%) | 2 (50%) | 0 (NA) | 1 (100%) | 0 (NA) | 0 (NA) | 0 (NA) |
| Tetracycline (T) | 11 (61.1%) | 19 (59.4%) | 4 (100%) | 0 (NA) | 1 (100%) | 1 (50%) | 2 (100%) | 0 (NA) |
| Cefepime (FEP) | 10 (55.6%) | 6 (18.8%) | 4 (100%) | 0 (NA) | 0 (NA) | 1 (50%) | 2 (100%) | 0 (NA) |
| Vancomycin (VA) | 18 (100%) | 32 (100%) | 4 (100%) | 3 (100%) | 1 (100%) | 2 (100%) | 2 (100%) | 2 (100%) |
| Trimethoprim-Sulfa (SXT) | 8 (44.4%) | 14 (43.8%) | 2 (50%) | 0 (NA) | 1 (100%) | 1 (50%) | 2 (100%) | 0 (NA) |
| Norfloxacin (NOR) | 6 (33.3%) | 5 (15.6%) | 0 (NA) | 0 (NA) | 1 (100%) | 0 (NA) | 1 (50%) | 0 (NA) |
| Amoxicillin Claviculenic Acid (AMC) | 6 (33.3%) | 6 (18.8%) | 1 (25%) | 0 (NA) | 0 (NA) | 1 (50%) | 2 (100%) | 2 (100%) |
| Ciprofloxacin (CIP) | 10 (55.6%) | 13 (40.6%) | 4 (100%) | 0 (NA) | 1 (100%) | 0 (NA) | 1 (50%) | 1 (50%) |
| Gentamicin (CN) | 9 (50%) | 4 (12.5%) | 0 (NA) | 0 (NA) | 0 (NA) | 1 (50%) | 0 (NA) | 1 (50%) |
| Erythromycin (E) | 18 (100%) | 32 (100%) | 4 (100%) | 3 (100%) | 1 (100%) | 2 (100%) | 2 (100%) | 2 (100%) |
| Oxacillin (OX) | 18 (100%) | 32 (100%) | 4 (100%) | 3 (100%) | 1 (100%) | 2 (100%) | 2 (100%) | 2 (100%) |
| Methicillin (ME) | 18 (100%) | 32 (100%) | 4 (100%) | 3 (100%) | 1 (100%) | 2 (100%) | 2 (100%) | 2 (100%) |

**Table 4. Prevalence and number of resistant isolates in this study.**

| Antimicrobial Agent | A. baumannii; 2 Isolates | E. faecalis; 2 Isolates | S. agalactiae; 1 Isolates | L. amnigena; 1 Isolates |
|---|---|---|---|---|
| | Pediatrics; 2 | Pregnant Women; 2 | Pregnant Women; 1 | Pediatrics; 1 |
| Penicillin (P) | 2 (100%) | 1 (50%) | 1 (100%) | 1 (100%) |
| Ampicillin (AMP) | 2 (100%) | 0 (NA) | 1 (100%) | 1 (100%) |
| Chloramphenicol (C) | 2 (100%) | 1 (50%) | 1 (100%) | 0 (NA) |
| Tetracycline (T) | 2 (100%) | 1 (50%) | 1 (100%) | 1 (100%) |
| Cefepime (FEP) | 1 (50%) | 2 (100%) | 1 (100%) | 0 (NA) |
| Vancomycin (VA) | 2 (100%) | 2 (100%) | 1 (100%) | 1 (100%) |
| Trimethoprim-Sulfa (SXT) | 0 (NA) | 2 (100%) | 0 (NA) | 0 (NA) |
| Norfloxacin (NOR) | 0 (NA) | 0 (NA) | 1 (100%) | 0 (NA) |
| Amoxicillin Clavulanic Acid (AMC) | 2 (100%) | 0 (NA) | 0 (NA) | 0 (NA) |
| Ciprofloxacin (CIP) | 0 (NA) | 1 (50%) | 0 (NA) | 1 (100%) |
| Gentamicin (CN) | 0 (NA) | 1 (50%) | 1 (100%) | 1 (100%) |
| Erythromycin (E) | 0 (NA) | 1 (50%) | 1 (100%) | 1 (100%) |
| Oxacillin (OX) | 2 (100%) | 2 (100%) | 0 (NA) | 1 (100%) |
| Methicillin (ME) | 2 (100%) | 2 (100%) | 1 (100%) | 1 (100%) |

Resistance levels of E. coli isolates varied according to age, in the case of Cefepime, Gentamicin, Norfloxacin, Amoxicillin-Clavulanic Acid and Ciprofloxacin, higher resistance was found among adults when compared to pediatrics.

observation was found in Nigeria (31.7%) [17], Croatia (62.6%) [18], Iran (74.6%) [19], Britain (65.1%) [20] and two studies from the USA (68 and 69%) [13,14]. The second most isolated bacteria was *K. pneumoniae* (10%), it was also the second most common in Nigeria (21.9%) and Iran (11.7%) [17,19]. This result indicates that *K. pneumoniae* is becoming more common as an etiological agent of UTI [17]. Other bacterial species were isolated at a lower rate which was consistent with other studies [20].

A high resistance to Amoxicillin-Clavulanic Acid antibiotic was found among *E. coli* isolates (40%), this is in keeping with findings in France where the resistance was also high

**Table 5. Multidrug resistance (MDR) values for all tested isolates in pregnant and pediatric patients.**

| Isolate | MDR Range | Mode | Aggregate MDR |
|---|---|---|---|
| *E. coli* | 0.36 - 0.93 | 0.64 | 0.62 |
| *Klebsiella* | 0.43 - 0.79 | 0.43, 0.71 | 0.60 |
| *Enterobacter* | 0.50 - 0.86 | NA | 0.68 |
| *Proteus* | 0.43 - 0.79 | 0.79 | 0.67 |
| *Lelliottia amnigena* | 0.65 | NA | 0.65 |
| *Acinetobacter* | 0.64 - 0.73 | NA | 0.68 |
| *Streptococcus* | 0.83 | NA | 0.83 |
| *Enterococcus faecalis* | 0.38 - 0.69 | NA | 0.54 |

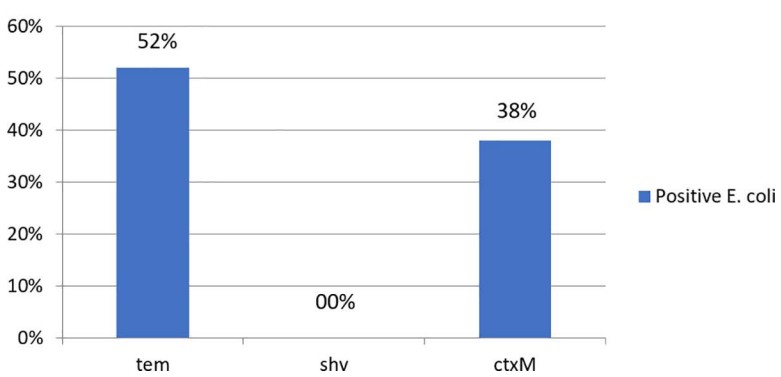

**Fig 2. The percentages of ES βL detection in *E. coli* isolates, the most prevalent ESβL gene was TEM followed by CTX-M, no SHV gene was detected.**

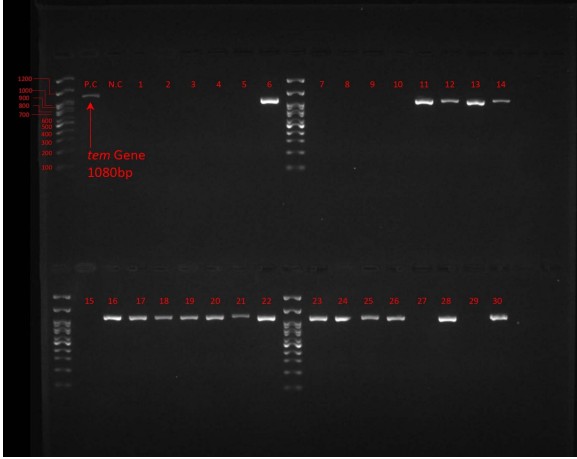

**Fig 3. Electrophoresis of TEM gene, 100 bp Ladder, Lane P.C: Positive Control, Lane N.C: Negative Control, Lane 1-30: Isolates.**

(29.7%) [21]. Resistance rates from other sources were generally lower, Croatia (3.01%) [18], USA 0.5% [22], in Denmark, Finland, Norway and Sweden (2.9%), and in UK and Ireland (3.7%) [23]. Resistance to drugs like Amoxicillin is of great importance, it means that it can't

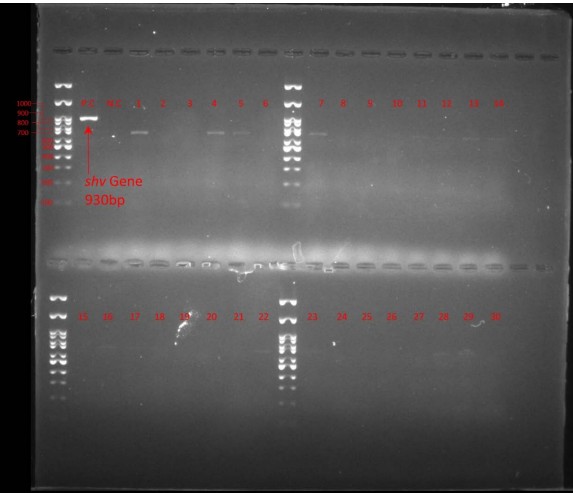

**Fig 4. Electrophoresis of SHV gene, 100 bp Ladder, Lane P.C: Positive Control, Lane N.C: Negative Control, Lane 1-30: Isolates.**

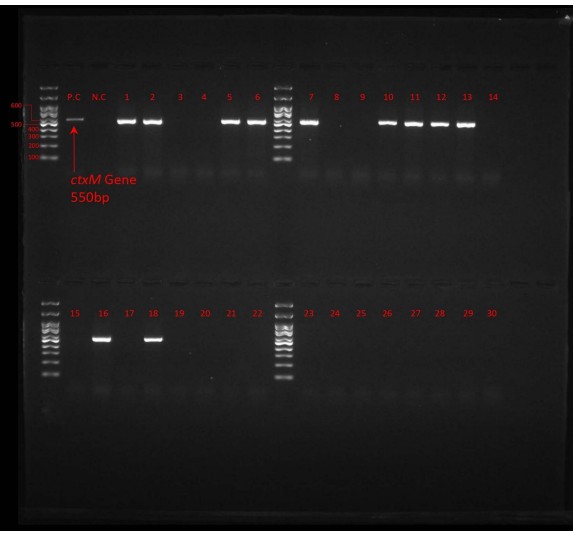

**Fig 5. Electrophoresis of CTX-M gene, 100 bp Ladder, Lane P.C: Positive Control, Lane N.C: Negative Control, Lane 1-30: Isolates.**

be used as an empirical therapy for UTI [18]. Trimethoprim-sulfa is used to treat uncomplicated urinary tract infections; the fact that almost all isolates had high resistance means that it also can't be used for UTI treatment [21,24,25].

Resistance to Ciprofloxacin was 48% among all *E. coli* isolates, worldwide resistance ranges from 8.4% to 92.9%, and resistance levels in common bacterial infections are increasing which indicates that effective antibiotics are decreasing [26].

A global trend of increase in antibiotic resistance has been shown worldwide. The main reason for this trend is the increase in antibiotic consumption [27]. Another reason for resistance is the high rate of antimicrobial usage by families, even in the absence of a prescription [19].

In the present study *E. coli* accounted for 71.4% of the UTI cases, the average MAR was 62% moreover, 52% of *E. coli* harbored TEM gene and 38% harbored CTX-M gene, 22% harbored both.

The main etiological agent of UTI is *E. coli* (accounting for 70–90% of all cases), due to the rapid spread of antimicrobial resistance especially ESβL-producing *E. coli*, treatment of patients has become more difficult [28,29]. The most common and important mechanism through which bacteria can become resistant against β-lactams is by expressing β-lactamases, for example extended-spectrum β-lactamases (ESβLs) [30].

Several studies have described the prevalence of ESβL-*E. coli* in UTI cases, in Qatar the prevalence of *E. coli* in UTI cases was 83% out of which 57.2% were ESβL-producing [31]. In Egypt, the prevalence of *E. coli* in UTI cases was 68.6% out of which 59.7% were ESβL-producing [32]. Although TEM was detected more than CTX-M in this study, global prevalence shifted towards CTX-M rather than TEM [33]. Nowadays, the bacterial resistance rate against β-lactams is increasing significantly and has become a common global health problem [34].

## Conclusion

In conclusion, gram-negative bacteria mainly *E. coli* was found to be as the most prevalent cause of urinary tract infections in pregnant women and pediatrics. Furthermore, pediatrics were more susceptible to urinary tract infections than pregnant women and resistance of UTI-causing gram-negative bacteria against gentamicin was generally low. High prevalence of multi-drug-resistant bacteria was found in this study, ESβL resistance in *E. coli* is mainly regulated by TEM and CTX-M genes. All these results highlight the need for regulating the use of antibiotics according to national and international guidelines and further investigate the emergence of MAR bacteria in certain populations. Furthermore, more research is needed in this field to find other therapeutic options and possible new antibiotic targets in the causative bacteria for these infections to decrease the complications of these infections in patients and their impact on the health care systems.

## Supporting information

**S1 Data. PCR Images.**
(ZIP)

**S2 Data. Results Summary sheet.**
(XLSX)

## Author contributions

**Data curation:** Leen Al Momani.

**Investigation:** Rama Al Titi.

**Methodology:** Ismail Malkawi.

**Resources:** Mohammad Al-Magableh.

**Supervision:** Waleed Al Momani.

**Writing – original draft:** Mohammad Al-Magableh, Ayah Elayan.

**Writing – review & editing:** Mohammad Al-Magableh.

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
