## [Decision Letter · Decision Letter 0]

4 Jul 2024

PONE-D-24-18526Extended spectrum-lactamase (ESBL)-producing E. coli causing urinary tract infection among adult and pediatric patients in public hospitals in northern JordanPLOS ONE

Dear Dr. al-magableh,

Thank you for submitting your manuscript to PLOS ONE. After careful consideration, we feel that it has merit but does not fully meet PLOS ONE’s publication criteria as it currently stands. Therefore, we invite you to submit a revised version of the manuscript that addresses the points raised during the review process.

We look forward to receiving your revised manuscript.

Kind regards,

Tebelay Dilnessa, MSc

Academic Editor

PLOS ONE

Journal Requirements:

https://en.wikipedia.org/wiki?curid=40114

https://www.sciencedirect.com/science/article/abs/pii/S0924857903002334?via%3Dihub

In your revision ensure you cite all your sources (including your own works), and quote or rephrase any duplicated text outside the methods section. Further consideration is dependent on these concerns being addressed.

Additional Editor Comments:

The paper raises a critical public health importance even though it was written carelessly. Totally, I share the two reviewers’ concern. At this time, I do not give you a point-by-point comment, rather thinking that the paper will be improved and to be written to the scientific standard, and PLOS ONE manuscript writing protocol I generously give you the chance to revise the paper extensively.The laboratory method should be written properly.Tables and figures descriptions should be well explanatory in terms of person, place and time.The citations were not correct and follow the protocol of PLOS ONE.Conclusion and/or recommendation was required.Figures should be prepared without description in tif and uploaded. The descriptions and legends should be placed within the text where you cited.The following should be added:**Funding, Availability of data and materials** , **Competing interests, Acknowledgments ** (if any), etc.

Reviewers' comments:

Reviewer's Responses to Questions

**Comments to the Author**

1. Is the manuscript technically sound, and do the data support the conclusions?

Reviewer #1: Partly

Reviewer #2: No

2. Has the statistical analysis been performed appropriately and rigorously? 

Reviewer #1: No

Reviewer #2: No

3. Have the authors made all data underlying the findings in their manuscript fully available?

Reviewer #1: Yes

Reviewer #2: No

4. Is the manuscript presented in an intelligible fashion and written in standard English?

Reviewer #1: No

Reviewer #2: No

5. Review Comments to the Author

Reviewer #1: Reviewer comment

Although the paper focuses on an important public heath problem and used advanced methods for gene detection it has major problems as follows:

Generally in the material and method section

The paper does not put the study area; study design and study period; source population; study population; inclusion and exclusion criteria of participants clearly.

The sample collection, bacterial identification and antibiotic susceptibility test are not described in detail and clear manner.

The reference guideline (CLSI or…) used for to say resistance, intermediate and sensitive is not mentioned.

The pcr condition, cycles, volumes …… totally not described.

The data analysis, quality control not included

In result section

The results written for each section is too shallow please elaborate more.

The tables for AST result (4 and 5) only described in percentage please include the number or frequency

The table title for table 4 it should be above the table

All the table titles and figure legends are too shallow. The tables and figures should be self explanatory.

No conclusion after the discussion

In general the manuscript has many problems which are difficult to comment line by line.

Reviewer #2: I have reviewed the manuscript entitled “Extended spectrum-lactamase (ESBL)-producing E. coli causing urinary tract infection among adult and pediatric patients in public hospitals in northern Jordan” submitted for possible publication in the journal “PlosOne”. Although the study provides a very important topic and addresses a global issue, it lacks the authentication of study. No doubt that the authors have done great efforts, but I will strongly suggest to add more data and the resubmit it. Because the results in the current form are very less and basic which needs to be improve. Some of my comments are:

1. Line 86-88: Time duration should be increase to get more number of isolates.

2. Line 87: Why only pregnant women and children? The authors should clarify and inclusion criteria and then amend the title accordingly.

3. Line 92: correct “°C”.

4. The bacterial identification criteria were not mentioned. Mention the exact criteria of bacterial identification or otherwise put the references.

5. Line 100: Why after the bacterial lawning, air dried the plates for 15 minutes? Any reference ?

6. Line 99: 0.5% McFarland standard.

7. Line 115-117: mention the bacterial species e.g., A. baumannii etc.

8. Line 119: here the authors has mentioned “Adults” but in methods “pregnant women” please clarify.

9. The authors have not mentioned about the CLSI or any other guidelines for AST testing which reduces the strength of study. Many of the drugs mentioned in table 4 and 5 are currently not using for all of the mentioned bacterial isolates. Please refer to any recent CLSI guidelines.

10. There is no conclusion section.

11. Figure 1: redundant with tables. Also according to the figure legend it should be age-wise groups.

12. Figure 2: redundant with tables.

6. PLOS authors have the option to publish the peer review history of their article (what does this mean? ). If published, this will include your full peer review and any attached files.

**Do you want your identity to be public for this peer review?** For information about this choice, including consent withdrawal, please see our Privacy Policy .

Reviewer #1: No

Reviewer #2: **Yes: ** Dr. Naveed Ahmed

---

## [Author Response · Author response to Decision Letter 1]

15 Sep 2024

Dear Tebelay Dilnessa PLOS ONE Editor

Thank you for the chance to edit our submission and re submit with the changes you and honourable reviewers suggested.

Please find response to the editors and reviews comments point by point.

1. Journal requirements

Up to our best knowledge, the manuscripts was made to meet PLOS ONE style requirements, including those for file naming and we tried to follow PLOS ONE style templates as much as we could and we adjusted any overlapping texts and cited any information that we added into our article. Up to our best knowledge, we included captions for the figures at the end of our manuscript, and we cited that in the texts when it was applicable.

To our best knowledge, the paper or the manuscripts writing has been improved to the scientific standard, and we tried to follow The PLOS ONE writing protocol. The laboratory method has been edited and written in a better way and the tables and figures description has been edited and explained in a better way. The citations, where edited and corrected to the best possible way and follow the protocol of PLOS ONE. Conclusions and recommendations were added and figures have been prepared and uploaded as a separate files. Funding and availability of data, conflict of interest and acknowledgement has been added to the manuscript.

Response to reviewers

Reviewer 1 comments. The manuscript and the results section have been edited and the methodology has been explained thoroughly and the sample size, date and place of the research has been added to the methodology section.

Statistical analysis has been reviewed and more information added to make it clearer and more understandable. All the Data is included in the manuscript and specific details and data will be available and provided upon request if needed. The manuscript has been edited and written in a better scientific way and has been reviewed multiple times. Grammar and typos has been checked and corrected up to our best knowledge.

Response to reviewer one comments the methodology section has been edited ,study area, study design and the period of the study has been added and inclusion and exclusion criteria as well. Reference guideline for antibiotic resistance and PCR conditions and data analysis has been included in the methodology section.

The results section has been edited to make it more clear and more understandable. The tables have been edited and more information added such as the number and the frequency, the tables and the figures legends have been edited and a conclusion section has been added as well.

In response to reviewer 2 comments, the results section has been expanded and explained in a better way. All the suggested corrections where corrected line by line. The bacterial identification criteria is added and CLSO guidelines has been included. A conclusion section has been added and the figures have been adjusted and added as a separate file.

Dear Editor

We have worked hard to improve our manuscript and adjusted all suggested comments to bring this manuscript up to the standard of your reputable journal, please kindly review our revised manuscript and we do hope that it will be accepted in your reputable journal.

Kind regards

---

## [Decision Letter · Decision Letter 1]

27 Nov 2024

PONE-D-24-18526R1Extended spectrum-lactamase (ESBL)-producing E. coli causing urinary tract infection among pregnant women and pediatric patients in public hospitals in northern JordanPLOS ONE

Dear Dr. al-magableh,

Thank you for submitting your manuscript to PLOS ONE. After careful consideration, we feel that it has merit but does not fully meet PLOS ONE’s publication criteria as it currently stands. Therefore, we invite you to submit a revised version of the manuscript that addresses the points raised during the review process.

We look forward to receiving your revised manuscript.

Kind regards,

Tebelay Dilnessa, MSc

Academic Editor

PLOS ONE

Additional Editor Comments:

The manuscript was significantly improved, but it requires a through revision.In the abstract, use at least the two versions at the same time for abbreviations, e.g, Urinary tract infections (UTI),…….There was no ‘objective’ of the study in the abstract partWhen, where and from whom the study was conducted? It should be included in both the abstract and the methods part.How could you collect the sample? What type of urine sample was collected?In the abstract and result, the absolute number (numerator and denominator) is needed together with the percentage. For example, in the abstract for *E. coli* A/B (71.4%).Your conclusion is explanation. Please make it conclusion, and try to decrease the numbers.All key words are abbreviations, please write their long formParagraphing was needed throughout the document, especially in the introduction and discussion part.It is better you organize the ‘Materials and Methods’ part as follows

Materials and Methods

Study area and setting

Study design and period

Study population

Sample size determination and sampling technique (if any)

Samples collection and bacterial isolation

Ethical Considerations

A separate ‘Ethical consideration’ statement is needed.Table 1 was not needed. You can write it as a statement. The antimicrobials used were ampicillin (AMP, 10µg), norfloxacin (NOR, 10µg), ……………..Line 123: **Bacterial isolates, ** not bacterial isolation and identificationLine 124: A sentence cannot start by number and abbreviation. ‘70 bacterial isolates were identified…..’ It is better written as, ‘Seventy bacterial isolates were identified…..’Line 150: The following should be below the tables. R: Resistant; S: Susceptible; I: Intermediate.Line 160: Table 6: MAR values for all tested isolates. Please write the long form of “MAR” and the description should be full in terms of person, place and time.Line 163: ……was the most frequent (n= 26, 52%) followed by ESβLCTXM (N= 19, 38%) (Figure 1 & 2).Critically revise citations of figures and tables.Funding, Availability of data and materials, Competing interests and Acknowledgments should be written in standard English.Tables are too poor and you have to prepare them properly.The references were not written properly. All references should be written correctly according to Vancouver style.Use the alignment ‘Justify’ for the text throughout the manuscriptThe reference writing was not correct and need revisionFollow the standard binomial nomenclature, italize journal name and the word ‘et al’Follow the guideline for manuscript writing protocol for PLoS One.

Reviewers' comments:

Reviewer's Responses to Questions

**Comments to the Author**

1. If the authors have adequately addressed your comments raised in a previous round of review and you feel that this manuscript is now acceptable for publication, you may indicate that here to bypass the “Comments to the Author” section, enter your conflict of interest statement in the “Confidential to Editor” section, and submit your "Accept" recommendation.

Reviewer #1: (No Response)

2. Is the manuscript technically sound, and do the data support the conclusions?

Reviewer #1: Partly

3. Has the statistical analysis been performed appropriately and rigorously? 

Reviewer #1: Yes

4. Have the authors made all data underlying the findings in their manuscript fully available?

Reviewer #1: Yes

5. Is the manuscript presented in an intelligible fashion and written in standard English?

Reviewer #1: No

6. Review Comments to the Author

Reviewer #1: Reviewer comment

The authors have addressed most of the previous reviewer comments; however, the manuscript still has many problems.

Line 1: insert symbol for beta before word lactamase

Line 23-24: remove this sentence “and is responsible for 60-80% of these infections.”

Line 24-25: replace this sentence “UTI is classified based on clinical presentation and diagnosis into upper UTI and lower UTI” with this: “based on the clinical presentation and diagnosis UTI classified into upper and lower UTI”

Line 32,33,39 ,126,127,133 and other parts of the manuscript write the scientific name of bacterial isolates correctly: species name is not started in capital letter: write Klebsiella pneumoniae correctly

"Line 40 states, 'Multiple antibiotic resistance.' What does it mean? Does it refer to 'multidrug resistant'?"

Line 41: “0.62” change it in to percentage

Line 37-44: The conclusion section in the abstract appears to present specific results rather than a conclusion. Please provide a general conclusion of the study based on your results.

I started assessing the manuscript line by line, but I couldn’t reach the end because it is full of errors

General comments

Some scientific name of bacterial isolates note written correctly.

There no definition or criteria for multidrug resistance isolates

Conclusion statement in the abstract and the conclusion section are completely different.

In the method section

The ethical statement is included in the study design section

Please list the names of the antibiotics used in a statement rather than in a table.

There no reference putted for the for CLSI in the method section

Please provide the detailed PCR conditions in a statement rather than in a table, starting from the initial denaturation to the final extension.

The author comparing the prevalence of bacterial isolates and antibiotic resistance results among different groups without including statistical significance values

In some cases, the author uses only the genus, while in others, species of isolates are used. See Tables 4, 5, 6, and others.

Finally, although this study used advanced methods like VITAKE and PCR, the authors need a well-experienced microbiologist to understand the results and rewrite the manuscript. Moreover, the author needs individuals for proofreading in English for grammar, spelling, coherence, and flow of sentence.

7. PLOS authors have the option to publish the peer review history of their article (what does this mean? ). If published, this will include your full peer review and any attached files.

**Do you want your identity to be public for this peer review?** For information about this choice, including consent withdrawal, please see our Privacy Policy .

Reviewer #1: No

---

## [Author Response · Author response to Decision Letter 2]

11 Dec 2024

Additional Editor Comments:

The editor's comments, and the reviewers’ comments are highly appreciated, and valued. Please find the response point by point and the changes made. We did our best effort to adjust and improve the quality of this article. Your insight and input is much appreciated.

• The manuscript was significantly improved, but it requires a through revision. Done with care.

• In the abstract, use at least the two versions at the same time for abbreviations, e.g, Urinary tract infections (UTI),……. Done.

• There was no ‘objective’ of the study in the abstract part. Added

• When, where and from whom the study was conducted? It should be included in both the abstract and the methods part. Added

• How could you collect the sample? What type of urine sample was collected? Added

• In the abstract and result, the absolute number (numerator and denominator) is needed together with the percentage. For example, in the abstract for E. coli A/B (71.4%). Done.

• Your conclusion is explanation. Please make it conclusion, and try to decrease the numbers. Done.

• All key words are abbreviations, please write their long form Done.

• Paragraphing was needed throughout the document, especially in the introduction and discussion part. Done.

• It is better you organize the ‘Materials and Methods’ part as follows Done.

Materials and Methods

Study area and setting

Study design and period

Study population

Sample size determination and sampling technique (if any)

Samples collection and bacterial isolation

•

•

•

Ethical Considerations

• A separate ‘Ethical consideration’ statement is needed. Will upload separately.

• Table 1 was not needed. You can write it as a statement. The antimicrobials used were ampicillin (AMP, 10µg), norfloxacin (NOR, 10µg), …………….. Removed. Table 1.

• Line 123: Bacterial isolates, not bacterial isolation and identification Done.

• Line 124: A sentence cannot start by number and abbreviation. ‘70 bacterial isolates were identified…..’ It is better written as, ‘Seventy bacterial isolates were identified…..’ Done.

• Line 150: The following should be below the tables. R: Resistant; S: Susceptible; I: Intermediate. Changed The title to resistance percentages.

• Line 160: Table 6: MAR values for all tested isolates. Please write the long form of “MAR” and the description should be full in terms of person, place and time. Done.

• Line 163: ……was the most frequent (n= 26, 52%) followed by ESβLCTXM (N= 19, 38%) (Figure 1 & 2). Done.

• Critically revise citations of figures and tables. Done.

• Funding, Availability of data and materials, Competing interests and Acknowledgments should be written in standard English. Done.

• Tables are too poor and you have to prepare them properly. Hopefully. Done. And adjusted.

• The references were not written properly. All references should be written correctly according to Vancouver style. Hopefully. Done. And adjusted.

• Use the alignment ‘Justify’ for the text throughout the manuscript Done.

•

• The reference writing was not correct and need revision Hopefully. Done. And adjusted.

•

• Follow the standard binomial nomenclature, italize journal name and the word ‘et al’ done

• Follow the guideline for manuscript writing protocol for PLoS One. Hopefully. Done. And adjusted.

•

Reviewers' comments:

Reviewer's Responses to Questions

Comments to the Author

1. If the authors have adequately addressed your comments raised in a previous round of review and you feel that this manuscript is now acceptable for publication, you may indicate that here to bypass the “Comments to the Author” section, enter your conflict of interest statement in the “Confidential to Editor” section, and submit your "Accept" recommendation.

Reviewer #1: (No Response)

2. Is the manuscript technically sound, and do the data support the conclusions?

The manuscript must describe a technically sound piece of scientific research with dataooypdd that supports the conclusions. Experiments must have been conducted rigorously, with appropriate controls, replication, and sample sizes. The conclusions must be drawn appropriately based on the data presented.

Reviewer #1: Partly

3. Has the statistical analysis been performed appropriately and rigorously?

Reviewer #1: Yes

4. Have the authors made all data underlying the findings in their manuscript fully available?

Reviewer #1: Yes

5. Is the manuscript presented in an intelligible fashion and written in standard English?

Reviewer #1: No

6. Review Comments to the Author

• Reviewer #1: Reviewer comment

The authors have addressed most of the previous reviewer comments; however, the manuscript still has many problems.

Line 1: insert symbol for beta before word lactamase Done.

Line 23-24: remove this sentence “and is responsible for 60-80% of these infections.” Done.

Line 24-25: replace this sentence “UTI is classified based on clinical presentation Done.

• and diagnosis into upper UTI and lower UTI” with this: “based on the clinical presentation and diagnosis UTI classified into upper and lower UTI” Done.

Line 32,33,39 ,126,127,133 and other parts of the manuscript write the scientific name of bacterial isolates correctly: species name is not started in capital letter: write Klebsiella pneumoniae correctly Done.

"Line 40 states, 'Multiple antibiotic resistance.' What does it mean? Does it refer to 'multidrug resistant'?" Done.

Line 41: “0.62” change it in to percentage Done.

Line 37-44: The conclusion section in the abstract appears to present specific Done.

results rather than a conclusion. Please provide a general conclusion of the study based on your results. Done.

I started assessing the manuscript line by line, but I couldn’t reach the end because it is full of errors

General comments

Some scientific name of bacterial isolates note written correctly. Done.

There no definition or criteria for multidrug resistance isolates Done.

Conclusion statement in the abstract and the conclusion section are completely different. Done.

In the method section

The ethical statement is included in the study design section Done.

Please list the names of the antibiotics used in a statement rather than in a table. Done.

There no reference putted for the for CLSI in the method section

Please provide the detailed PCR conditions in a statement rather than in a table, starting from the initial denaturation to the final extension.

The author comparing the prevalence of bacterial isolates and antibiotic resistance results among different groups without including statistical significance values

In some cases, the author uses only the genus, while in others, species of isolates are used. See Tables 4, 5, 6, and others. Done.

Finally, although this study used advanced methods like VITAKE and PCR, the authors need a well-experienced microbiologist to understand the results and rewrite the manuscript. Moreover, the author needs individuals for proofreading in English for grammar, spelling, coherence, and flow of sentence. Hopefully done and improved

7. PLOS authors have the option to publish the peer review history of their article (what does this mean?). If published, this will include your full peer review and any attached files.

Do you want your identity to be public for this peer review? For information about this choice, including consent withdrawal, please see our Privacy Policy.

Reviewer #1: No

---

## [Decision Letter · Decision Letter 2]

30 Dec 2024

PONE-D-24-18526R2Extended spectrum β-lactamase (ESβL)-producing E. coli causing urinary tract infection among pregnant women and pediatric patients in public hospitals in northern JordanPLOS ONE

Dear Dr. al-magableh,

Thank you for submitting your manuscript to PLOS ONE. After careful consideration, we feel that it has merit but does not fully meet PLOS ONE’s publication criteria as it currently stands. Therefore, we invite you to submit a revised version of the manuscript that addresses the points raised during the review process.

We look forward to receiving your revised manuscript.

Kind regards,

Tebelay Dilnessa, MSc

Academic Editor

PLOS ONE

Additional Editor Comments (if provided):

- It is expected the author critically see the comments and incorporate or respond to the comments.

- Follow critically the submission guideline, specially the declariation part. It should be formal as follows.

**Funding**

This research received no specific fund.

**Availability of data and materials**

All data information generated in this study was included in the manuscript.

**Competing interests**

Authors declare that they have no competing interests.

Reviewers' comments:

Reviewer's Responses to Questions

**Comments to the Author**

1. If the authors have adequately addressed your comments raised in a previous round of review and you feel that this manuscript is now acceptable for publication, you may indicate that here to bypass the “Comments to the Author” section, enter your conflict of interest statement in the “Confidential to Editor” section, and submit your "Accept" recommendation.

Reviewer #1: (No Response)

Reviewer #3: (No Response)

2. Is the manuscript technically sound, and do the data support the conclusions?

Reviewer #1: Partly

Reviewer #3: Partly

3. Has the statistical analysis been performed appropriately and rigorously? 

Reviewer #1: N/A

Reviewer #3: Yes

4. Have the authors made all data underlying the findings in their manuscript fully available?

Reviewer #1: Yes

Reviewer #3: Yes

5. Is the manuscript presented in an intelligible fashion and written in standard English?

Reviewer #1: No

Reviewer #3: No

6. Review Comments to the Author

Reviewer #1: Although the authors addressed some of my previous comments, the manuscript still requires significant revisions. Many sections need to be rewritten by a medical microbiologist to ensure accuracy and clarity. Below are some comments; however, it is challenging to fully assess the manuscript due to its current state.

• The gel picture is snipped one put picture that show the ends of the gel

• Figure2 is not labeled like ladder (label as 100bp, 200bp, 300bp… up to the target gene)

• Genes detected not labeled in the figure (look other publications)

• Discussion should start by explaining your work not others

• Line 185-191 shows the comparison of others work without even indicating your work

• The result of molecular work not well written both in the result and discussion section

• Figure 1 only shows result of two genes what about the other two even if they were not detected they should be indicated as zero in frequency or percentage

• Table titles are not included the study area and time.

• The term "multiple antibiotic resistance" is incorrect and should be replaced with "multidrug resistance (MDR)." Please use the provided reference for identifying and counting MDR isolates.

o (Magiorakos A-P, Srinivasan A, Carey RB, Carmeli Y, Falagas M, Giske C, et al. Multidrug-resistant, extensively drug-resistant and pandrug-resistant bacteria: an international expert proposal for interim standard definitions for acquired resistance. Clinical microbiology and infection. 2012;18(3):268-81.)

Reviewer #3: Abstract

1. Background: The sentence "Based on the clinical presentation and diagnosis, UTI is classified into upper and lower UTI" should be removed. Instead, focus on the main gaps identified that led to the selection of the study title.

2. Methodology: The methodology section is underdeveloped. Please expand it to include the total number of study participants, the sampling technique used, and the methods for data entry and analysis.

3. Results: Acceptable as is.

4. Conclusion: The conclusion section is fine, but it is recommended to remove any percentage descriptions.

Introduction

1. The introduction does not provide a clear explanation of extended spectrum β-lactamase (ESβL)-producing E. coli in relation to urinary tract infections (UTIs) among pregnant women and pediatric patients. Revise to make this connection clearer.

2. Clarify how the study title was selected. What specific gaps or issues prompted this title?

3. Provide scientific reasoning or evidence to explain the occurrence of UTI relapse in the study population.

4. Use updated references to strengthen the introduction.

5. Overall, rewrite the introduction to improve clarity and focus on the study's objectives and significance.

Methods

1. The ethical considerations should not be included under "Study Setting and Population". Please move it to a separate section for clarity.

2. Key sections are missing, including source and study population, sample size determination, sampling technique, and data analysis methods. Include a description of the data collection procedure, along with details on data entry and data analysis.

7. PLOS authors have the option to publish the peer review history of their article (what does this mean? ). If published, this will include your full peer review and any attached files.

**Do you want your identity to be public for this peer review?** For information about this choice, including consent withdrawal, please see our Privacy Policy .

Reviewer #1: No

Reviewer #3: No

---

## [Author Response · Author response to Decision Letter 3]

21 Jan 2025

The editor's comments, and the reviewers’ comments are highly appreciated, and valued. Please find the response point by point and the changes made. We did our best effort to adjust and improve the quality of this article. Your insight and input is much appreciated.

Comment Reply

Regarding reviewing the revision by a microbiologist One of the main authors who is a known microbiologist has reviewed and approved the manuscripts and helped in re writing it, with thanks

The gel picture is snipped one put picture that show the ends of the gel All figures were updated, Figures 1 to 5.

Figure2 is not labeled like ladder (label as 100bp, 200bp, 300bp… up to the target gene) All figures were updated, Figures 1 to 5.

Genes detected not labeled in the figure (look other publications) All figures were updated, Figures 1 to 5.

Discussion should start by explaining your work not others Updated

Line 185-191 shows the comparison of others work without even indicating your work Added “ In this study, the occurrence of UTI in children was higher than adults (60% and 40% respectively).”

The result of molecular work not well written both in the result and discussion section Changed

Figure 1 only shows result of two genes what about the other two even if they were not detected they should be indicated as zero in frequency or percentage Percentages were Added

Table titles are not included the study area and time. See methodology section for study area and time

The term "multiple antibiotic resistance" is incorrect and should be replaced with "multidrug resistance (MDR)." Please use the provided reference for identifying and counting MDR isolates. Changed

Reviewer 1

Comment Reply

Regarding reviewing the revision by a microbiologist One of the main authors who is a known microbiologist has reviewed and approved the manuscripts and helped in re writing it, with thanks

The gel picture is snipped one put picture that show the ends of the gel All figures were updated, Figures 1 to 5.

Figure2 is not labeled like ladder (label as 100bp, 200bp, 300bp… up to the target gene) All figures were updated, Figures 1 to 5.

Genes detected not labeled in the figure (look other publications) All figures were updated, Figures 1 to 5.

Discussion should start by explaining your work not others Updated

Line 185-191 shows the comparison of others work without even indicating your work Added “ In this study, the occurrence of UTI in children was higher than adults (60% and 40% respectively).”

The result of molecular work not well written both in the result and discussion section Changed

Figure 1 only shows result of two genes what about the other two even if they were not detected they should be indicated as zero in frequency or percentage Percentages were Added

Table titles are not included the study area and time. See methodology section for study area and time

The term "multiple antibiotic resistance" is incorrect and should be replaced with "multidrug resistance (MDR)." Please use the provided reference for identifying and counting MDR isolates. Changed

Reviewer 2

Comment Reply

Background: The sentence "Based on the clinical presentation and diagnosis, UTI is classified into upper and lower UTI" should be removed. Instead, focus on the main gaps identified that led to the selection of the study title. This was requested by the first reviewer

Methodology: The methodology section is underdeveloped. Please expand it to include the total number of study participants, the sampling technique used, and the methods for data entry and analysis. updated

Results: Acceptable as is. Thank you

Conclusion: The conclusion section is fine, but it is recommended to remove any percentage descriptions. done

The introduction does not provide a clear explanation of extended spectrum β-lactamase (ESβL)-producing E. coli in relation to urinary tract infections (UTIs) among pregnant women and pediatric patients. Revise to make this connection clearer. See lines 75-78 in the introduction with thanks

Clarify how the study title was selected. What specific gaps or issues prompted this title? See methodology please

Provide scientific reasoning or evidence to explain the occurrence of UTI relapse in the study population. See conclusion please

Use updated references to strengthen the introduction. Checked and done thanks

Overall, rewrite the introduction to improve clarity and focus on the study's objectives and significance. Made changes thanks for the comment

The ethical considerations should not be included under "Study Setting and Population". Please move it to a separate section for clarity. Changed

Key sections are missing, including source and study population, sample size determination, sampling technique, and data analysis methods. Include a description of the data collection procedure, along with details on data entry and data analysis. Sampling technique is mentioned, line 101 “Samples were collected from each patient using the midstream clean-catch technique [8]”

---

## [Decision Letter · Decision Letter 3]

23 Jan 2025

PONE-D-24-18526R3Extended spectrum β-lactamase (ESβL)-producing E. coli causing urinary tract infection among pregnant women and pediatric patients in public hospitals in northern JordanPLOS ONE

Dear Dr. al-magableh,

Thank you for submitting your manuscript to PLOS ONE. After careful consideration, we feel that it has merit but does not fully meet PLOS ONE’s publication criteria as it currently stands. Therefore, we invite you to submit a revised version of the manuscript that addresses the points raised during the review process.

We look forward to receiving your revised manuscript.

Kind regards,

Tebelay Dilnessa, MSc

Academic Editor

PLOS ONE

Journal Requirements:

Additional Editor Comments:

Line 27: Methods: This study was conducted in Public-Teaching Hospitals in……Why you use outdated CLSI guidelines (ref. 32 and 33) than updated CLSI, 2023 and 2024 guidelinesThese must be changed by updated one. ‘32. CLSI. Performance Standards for Antimicrobial Susceptibility Testing. 27th ed. CLSI supplement M100. Wayne, PA: Clinical and Laboratory Standards Institute; 2017; 33. CLSI. Performance Standards for Antimicrobial Susceptibility Testing. 23th ed. CLSI supplement M100. Wayne, PA: Clinical and Laboratory Standards Institute; 2013’.Tables and figures descriptions should be well explanatory in terms of person, place and time.Figure 2 should be prepared without title/figure description (It said Figure 1: Detection of ESBL in *E. coli* isolates. Even the figure is number 2, but you said Figure 1.You did not give response to the editor. I repeatedly told you that you have to follow the journal manuscript writing protocol.

Reviewers' comments:

Reviewer's Responses to Questions

**Comments to the Author**

1. If the authors have adequately addressed your comments raised in a previous round of review and you feel that this manuscript is now acceptable for publication, you may indicate that here to bypass the “Comments to the Author” section, enter your conflict of interest statement in the “Confidential to Editor” section, and submit your "Accept" recommendation.

Reviewer #3: (No Response)

2. Is the manuscript technically sound, and do the data support the conclusions?

Reviewer #3: Yes

3. Has the statistical analysis been performed appropriately and rigorously? 

Reviewer #3: Yes

4. Have the authors made all data underlying the findings in their manuscript fully available?

Reviewer #3: (No Response)

5. Is the manuscript presented in an intelligible fashion and written in standard English?

Reviewer #3: Yes

6. Review Comments to the Author

Reviewer #3: (No Response)

7. PLOS authors have the option to publish the peer review history of their article (what does this mean? ). If published, this will include your full peer review and any attached files.

**Do you want your identity to be public for this peer review?** For information about this choice, including consent withdrawal, please see our Privacy Policy .

Reviewer #3: **Yes: ** Abebe Fenta (MSc in Medical parasitology & vector control, MPH in Epidemiology)

---

## [Author Response · Author response to Decision Letter 4]

11 Feb 2025

Dear Editor.

The editor's comments and the reviewers’ comments are highly appreciated and valued. Please find the response point by point and the changes made. We did our best effort to adjust and improve the quality of this article. Your insight and input is much appreciated.

Editors’ comments

Journal Requirements:

Please review your reference list to ensure that it is complete and correct. If you have cited papers that have been retracted, please include the rationale for doing so in the manuscript text, or remove these references and replace them with relevant current references. Any changes to the reference list should be mentioned in the rebuttal letter that accompanies your revised manuscript. If you need to cite a retracted article, indicate the article’s retracted status in the References list and also include a citation and full reference for the retraction notice. Dear editor, I have gone through the references one by one and updated the reference list, I did not find any retracted articles, I have updated the CDC and WHO references to more recent ones and I have noticed the list was shifted in the old manuscript version by 1-2 numbers by mistake, I do apologize for that mistake, it was rectified and double checked. All the references are correct and numbered correctly.

Additional Editor Comments:

• Line 27: Methods: This study was conducted in Public-Teaching Hospitals in…… corrected

• Why you use outdated CLSI guidelines (ref. 32 and 33) than updated CLSI, 2023 and 2024 guidelines. Updated as advised

• These must be changed by updated one. ‘32. CLSI. Performance Standards for Antimicrobial Susceptibility Testing. 27th ed. CLSI supplement M100. Wayne, PA: Clinical and Laboratory Standards Institute; 2017; 33. CLSI. Performance Standards for Antimicrobial Susceptibility Testing. 23th ed. CLSI supplement M100. Wayne, PA: Clinical and Laboratory Standards Institute; 2013’. Done as advised much appreciated

• Tables and figures descriptions should be well explanatory in terms of person, place and time. Went through them again and hopefully changed to make it more clear.

• Figure 2 should be prepared without title/figure description (It said Figure 1: Detection of ESBL in E. coli isolates. Even the figure is number 2, but you said Figure 1. Corrected, thank you

• You did not give response to the editor. I repeatedly told you that you have to follow the journal manuscript writing protocol. Dear Editor. I went through the Journal publication guidelines Section by section. I've tried my best to make it compatible with the journal publication policies. Including headings, subheadings, sectioning and referencing. Hopefully, I made all the changes that is requested by the editor.

Reviewers comments,

Thank you for your response and for revising your manuscript based on my previous comments.

However, some critical issues have not been fully addressed in the revised manuscript. Please

make the necessary modifications based on the following comments:

Abstract

Background: The background lacks sufficient detail. The sentence "Based on clinical

presentation and diagnosis, UTI is classified into upper and lower UTI" should be removed.

Focus instead on the identified gaps in existing research or clinical practice that led to the study’s

title. Dear reviewer, the sentence specified has been removed. And the structure of the abstract. Has been modified to clarify the knowledge gaps.

Introduction: The introduction needs more clarity regarding the specific gaps that led to the

selection of this study title. Clearly state the issues or knowledge gaps that prompted the research

question and title. The introduction has been modified and a sentences has been added to make it more clear and to identify specific gaps that led to the selection of this study highlighted in the revised manuscript.

Methods:

•�Sample Size Determination: The rationale for selecting 70 isolates is unclear. Please

explain why 70 patients (isolates) were chosen and whether any statistical methods were

used for sample size calculation. The sample size determination was determined by the number of samples we could collect in the duration of the study that fits the criteria of inclusion. All patients (pregnant women and children) who met the criteria of getting UTI’s (had symptoms and culture resulted in a bacterial growth) were included in the study, seventy UTI cases were covered in this study. This was added to the methods section

•�Data Entry and Analysis: Specify the software used for data entry and analysis (e.g.,

EpiData, SPSS, Stata). Provide details of the analysis methods to ensure the results are

valid.The study was carried out during the period from the 1st of March 2024 to the 5th of April 2024 and all analysis was done on Microsoft Excel.

•�Inclusion and Exclusion Criteria: The criteria for participant inclusion and exclusion need

clarification. Specify the symptoms required for inclusion and whether bacterial growth

was from study participants or existing data. The sample size determination was determined by the number of samples we could collect in the duration of the study that fits the criteria of inclusion. All patients (pregnant women and children) who met the criteria of getting UTI’s (had symptoms and culture resulted in a bacterial growth) were included in the study, seventy UTI cases were covered in this study. This was added to the methods section

•�Patient Population: Clarify the source of the 70 isolates (e.g., hospital patients, outpatient

clinics) and describe the population from which they were collected. A hospital-based prospective study was carried out in public-teaching Hospitals in northern Jordan with a total capacity of 400 beds (Badieah and Rahma Hospitals). 70 urine samples were collected from pediatrics and female outpatients. The study was carried out during the period from the 1st of March 2024 to the 5th of April 2024 and all analysis was done on Microsoft Excel. This aws added to the methods section

Comment Response

The background lacks sufficient detail. The sentence "Based on clinical

presentation and diagnosis, UTI is classified into upper and lower UTI" should be removed. Focus instead on the identified gaps in existing research or clinical practice that led to the study’s

title Done, Removed and More info were added

The introduction needs more clarity regarding the specific gaps that led to the

selection of this study title. Clearly state the issues or knowledge gaps that prompted the research

question and title. Added Highlighted in Green

The rationale for selecting 70 isolates is unclear. Please

explain why 70 patients (isolates) were chosen and whether any statistical methods were

used for sample size calculation. A hospital-based prospective study was carried out in public-teaching Hospitals in northern Jordan with a total capacity of 400 beds (Badieah and Rahma Hospitals). 70 urine samples were collected from pediatrics and female outpatients. The study was carried out during the period from the 1st of March 2024 to the 5th of April 2024 and all analysis was done on Microsoft Excel.

Specify the software used for data entry and analysis (e.g.,

EpiData, SPSS, Stata). Provide details of the analysis methods to ensure the results are

valid. Added Highlighted in Green

“and all analysis was done on Microsoft Excel”

The criteria for participant inclusion and exclusion need

clarification. Specify the symptoms required for inclusion and whether bacterial growth

was from study participants or existing data. The sample size determination was determined by the number of samples we could collect in the duration of the study that fits the criteria of inclusion. All patients (pregnant women and children) who met the criteria of getting UTI’s (had symptoms and culture resulted in a bacterial growth) were included in the study, seventy UTI cases were covered in this study. This was added to the methods section

clarify the source of the 70 isolates (e.g., hospital patients, outpatient

clinics) and describe the population from which they were collected. Added Highlighted in Green

Line 27: Methods: This study was conducted in Public-Teaching Hospitals in…… Changed

Why you use outdated CLSI guidelines (ref. 32 and 33) than updated CLSI Updated to CLSI 2024

Tables and figures descriptions should be well explanatory in terms of person, place and time.

Figure 2 should be prepared without title/figure description (It said Figure 1: Detection of ESBL in E. coli isolates. Even the figure is number 2, but you said Figure 1. Figure 1 is the electrophoresis image

Figure 2 is the chart

It now says Figure 2: Detection of ESBL in E. coli isolates.

---

## [Decision Letter · Decision Letter 4]

18 Feb 2025

Extended spectrum β-lactamase (ESβL)-producing E. coli causing urinary tract infection among pregnant women and pediatric patients in public hospitals in northern Jordan

PONE-D-24-18526R4

Dear Dr. al-magableh,

We’re pleased to inform you that your manuscript has been judged scientifically suitable for publication and will be formally accepted for publication once it meets all outstanding technical requirements.

Kind regards,

Tebelay Dilnessa, MSc

Academic Editor

PLOS ONE

Additional Editor Comments (optional):

-The descreption of figures should be also included below the reference lists.

Reviewers' comments:

Reviewer's Responses to Questions

**Comments to the Author**

1. If the authors have adequately addressed your comments raised in a previous round of review and you feel that this manuscript is now acceptable for publication, you may indicate that here to bypass the “Comments to the Author” section, enter your conflict of interest statement in the “Confidential to Editor” section, and submit your "Accept" recommendation.

Reviewer #3: All comments have been addressed

2. Is the manuscript technically sound, and do the data support the conclusions?

Reviewer #3: Yes

3. Has the statistical analysis been performed appropriately and rigorously? 

Reviewer #3: Yes

4. Have the authors made all data underlying the findings in their manuscript fully available?

Reviewer #3: Yes

5. Is the manuscript presented in an intelligible fashion and written in standard English?

Reviewer #3: Yes

6. Review Comments to the Author

Reviewer #3: All my concerns were addressed. Dear author l, please check the similarities from previous researchers through plagiarism checker.

7. PLOS authors have the option to publish the peer review history of their article (what does this mean? ). If published, this will include your full peer review and any attached files.

**Do you want your identity to be public for this peer review?** For information about this choice, including consent withdrawal, please see our Privacy Policy .

Reviewer #3: No

---

## [Editor Report · Acceptance letter]

PONE-D-24-18526R4

PLOS ONE

Dear Dr. Al-Magableh,

I'm pleased to inform you that your manuscript has been deemed suitable for publication in PLOS ONE. Congratulations! Your manuscript is now being handed over to our production team.

Kind regards,

on behalf of

Dr. Tebelay Dilnessa

Academic Editor

PLOS ONE